

# Uncertainties in Forecasts of Winter Storm Losses

Tobias Pardowitz[1,2], Robert Osinski[3], Tim Kruschke[4], Uwe Ulbrich[2]

[1] Hans Ertel Centre for Weather Research, Optimal Use of Weather Forecast Branch
[2] Freie Universität Berlin, Institute of Meteorology, Carl-Heinrich-Becker Weg 6-10, 12165 Berlin, Germany
[3] CNRM/GAME, Météo-France and CNRS, GMME/MICADO, 42 avenue Gustave Coriolis, 31057 Toulouse, France
[4] GEOMAR Helmholtz Centre for Ocean Research Kiel, Düsternbrooker Weg 20, 24105 Kiel, Germany

*Correspondence to*: Tobias Pardowitz (tobias.pardowitz@met.fu-berlin.de)

**Abstract.** This paper describes an approach to derive probabilistic predictions of local winter storm damage occurrences from a global medium-range ensemble prediction system (EPS). Predictions of storm damage occurrences are subject to large
uncertainty due to meteorological forecast uncertainty (typically addressed by means of ensemble predictions) and uncertainties in modelling weather impacts. The latter uncertainty arises from the fact that local vulnerabilities are not known in sufficient detail to allow for a deterministic assessment of damages. Thus to estimate the damage model uncertainty, a statistical model based on logistic regression analysis is employed, relating meteorological analyses to historical damage records. A quantification of the two individual contributions to the total forecast uncertainty is achieved by neglecting
individual uncertainty sources and analysing resulting predictions. Results show an increase in forecast skill if both meteorological and damage occurrence uncertainties are taken into account. It is demonstrated that skilful predictions on district level are possible on lead times of several days. Skill is increased through the application of a proper ensemble calibration method, extending the range of lead times for which skilful damage predictions can be made.

## 1 Introduction

Severe weather events and in particular severe winter storm events cause a major share of economic losses due to natural disasters in Europe and in Germany (MunichRe, 2007; MunichRe, 2012; MunichRe, 2013) and regularly cause a number of human fatalities.

To prevent human fatalities and reduce property losses caused by natural disasters, national and regional civil protection agencies need to be supported by effective weather warning systems. Within the Sendai framework for disaster risk reduction
(UNISDR, 2015), it has been stated, that for an effective disaster risk reduction an understanding of natural risks and their impacts is needed including all aspects of disasters such as vulnerability, capacity and exposure. With such understanding and, if possible, the ability to model the impacts of severe weather events improved warning systems should be designed, supporting decision making processes for tasks of civil protection agencies.

The modelling of winter storm damages in Germany has been carried out in a number of recent studies, including both
deterministic approaches (Klawa and Ulbrich, 2003; Heneka and Ruck, 2008; Donat et al. 2011) as well as probabilistic approaches (Heneka and Hofherr, 2010; Prahl et al., 2012). These storm damage models provide means to translate observed



or modelled wind speeds into local damage or loss ratios (i.e. losses normalized with the local sum of insured values). Depending on data availability these models include a regionally specific parameter estimation to describe differences in local vulnerabilities resulting e.g. from local differences in building characteristics (compare e.g. Donat et al., 2011). Rather than aiming at a quantitative model for predictions of loss ratios, here we employ a simple logistic regression model, aiming at the prediction of exceedance probabilities for defined loss thresholds. This model is similar to the first modelling step of the damage model described in Prahl et. al. (2012).

In giving an estimate of the inherent uncertainty in the relationship between wind and damage, this allows us to quantify the statistical model uncertainties arising in the impact-modelling step. The second major source for uncertainty in storm impact predictions arises from meteorological forecast uncertainties. The latter uncertainty is commonly addressed by means of ensemble prediction systems (Palmer, 2000; Leutbecher and Palmer, 2008; Slingo and Palmer, 2011) which is why we base our study on the medium range ensemble prediction system operationally run at the European Centre for Medium-Range Weather Forecasts (ECMWF; Palmer et al., 2007).

Our approach thus allows us to address and quantitatively compare the two main uncertainties arising in the modelling chain: meteorological forecast uncertainty and impact model uncertainty. In particular, we study the effect of neglecting uncertainty information, as commonly done when interpreting the ensemble mean of a forecast ensemble or applying a simple deterministic damage model neglecting the respective uncertainty.

The aim of this paper is to demonstrate the benefit of a fully probabilistic approach predicting storm damages, which can form the basis for the design of risk based warning tools. We furthermore aim at demonstrating the benefit (in terms of forecast skill) of an explicit and full treatment of the involved uncertainties within the modelling chain.

We structured the paper as follows. Chapter 2 describes the utilized data sources. The methodology, particularly the full modelling chain is described in chapter 3, including the verification methodology applied. Verification results are presented in chapter 4, followed by a discussion and conclusion in chapter 5.

## 2 Data

### 2.1 Insurance Loss Data

Insurance data on losses to residential buildings were provided by the German insurance association (Gesamtverband der Deutschen Versicherungswirtschaft e.V., GDV). These comprise daily data on administrative district level, with areas ranging from about 40 km² for urban municipalities ("Kreisfreie Städte") to about 3000 km² for rural districts ("Landkreise"). In contrast to pointwise measurements from meteorological stations, the available insurance data represent measurements with an area-wide coverage of windstorm and thunderstorm losses making it most valuable for various weather impact studies. The data however contains some limitations and uncertainties that need to be kept in mind. Uncertainties in daily losses arise from the fact that the exact time of loss occurrence is indistinct in some cases, especially if an event has occurred at night. Furthermore, the area representativeness implies a dependence of losses on the local building stocks that needs to be taken into



account. To gain data comparable amongst districts it is thus necessary to consider relative values i.e. losses standardized by the total amount of insured values (insured sum) in the specific district. Commonly used by insurances is the term loss ratio which denotes the loss (in €) divided by the insured sum (in thousand €) which is thus specified in ‰ (=1€/1000€). Besides ensuring spatial homogenization, the consideration of relative losses removes temporal inhomogeneity's resulting e.g. from

the growth of values or inflation. On district level, the GDV recorded losses on residential buildings arising from storm and hail events (covered by the "Verbundene Wohngebäude Versicherung", VGV) for the period 1997-2007. Here we consider the winter half year only (October through March). For these months, damages are almost exclusively caused by windstorms. However, in a few cases damages are due to hail which might take place in the vicinity of a storm's cold front (hail was observed e.g. in the case of Kyrill; compare Fink et al., (2009)). Since these hail induced damages cannot be systematically

separated in our analysis, this poses another uncertainty that needs to be reflected in the (probabilistic) relationship between local winds and resulting damages.

## 2.2 COSMO-EU Analyses

For training of the probabilistic storm damage model, analyses from the operational assimilation cycle for the COSMO-EU model (Schulz and Schättler, 2014) are employed. As a specific configuration of the non-hydrostatic COSMO-Model (Rockel

et al., 2008; Doms, 2011), COSMO-EU is operationally run at DWD covering the European domain in a resolution of 7km, using 40 vertical levels with the lowest level 10 meter above ground. Forecasts are operationally initialized 6 hourly (00, 06, 12 and 18 UTC) and performed for up to +78 hours. The COSMO-EU assimilation scheme (based on a nudging methodology) is performed 3 hourly (00, 03, 06, …, 21 UTC) and analysis files are written every hour. Here we use hourly wind gusts in 10m height, which are extracted for each hour from the latest available analysis run. These are finally used to calculate daily

maxima of wind gusts in 10 m height. The COSMO-EU analyses are available for the period 2006-2011.

## 2.3 ECMWF-EPS Forecasts

ECMWF is operationally running its Ensemble Prediction System (EPS) since 1992 (Molteni et al., 1996). This system is based on the same numerical weather prediction (NWP) model that is used for the deterministic weather forecast, the Integrated Forecasting System (IFS). However, in the ensemble prediction mode it is employed with a coarser vertical and horizontal

resolution. The latter has been increased from an initial resolution of T63 (~200km) over $T_L159$ (~120km; changed in 12/1996), $T_L255$ (~80km; 11/2000), and $T_L399$ (~50km; 02/2006) to the current resolution of $T_L639$ (~32km; since 01/2010). To generate the ensemble the method of singular vectors (Palmer et al., 1998; Leutbecher and Palmer, 2008) is used to perturb the initial conditions. Initially 32 ensemble members were produced. Since December 1996 this number was increased to 50 members. One additional control forecast is calculated using the same (unperturbed) initial conditions as the deterministic run, but

employing the coarser resolution of the EPS. Additionally, stochastic perturbations of the model physics were introduced in October 1998 (Buizza et al., 1999; Palmer et al., 2009).



For the current study we use the 6-hourly output of instantaneous 10 m wind speed of the 50 perturbed ensemble members operationally produced between 11/2000 and 01/2010 (in $T_L255$ and $T_L399$ resolution) as input for a statistical downscaling. Each forecast is integrated over 15 days, but the horizontal resolution is reduced after forecast day 10. We thus confine all further processing and analyses to the first ten forecast days of constant resolution throughout the respective integration.

## 3 Methodology

### 3.1 Statistical Downscaling of the ECMWF-EPS

Within the COSMO-EU-domain, the global ECMWF-EPS forecasts were statistically downscaled to the fine COSMO-EU-resolution of approx. 7 km, following the approach developed by Kruschke (2015). The basic concept of this downscaling procedure is a Multiple Linear Regression approach quantifying the relationship of fine scale surface gusts to the coarse scale (instantaneous) surface winds given by the respective ECMWF-EPS forecast.

For each COSMO-EU grid-box (436.905 in total) an individual statistical model, i.e. a regression equation is established. This is done by objectively choosing skilful predictors from a given set of potential predictors. Essentially, these potential predictors are the EPS surface wind components and wind magnitudes scaled by the respective climatological 98[th] percentile (to achieve homogenisation with respect to orographic effects) and subsequently interpolated (first-order conservative) to the coarser of the analysed EPS resolutions, that is TL159. More specifically, for each individual COSMO-EU grid box the scaled and interpolated instantaneous 6-hourly surface wind components and magnitudes at EPS grid boxes within a radius of 300 km (calculated between respective COSMO-EU and EPS grid box centres) as well as the squared values of these parameters are used to predict 6-hourly (temporal window centred over timestep of instantaneous predictors) maximum 10m-gusts at the respective COSMO-EU grid box. Scaling and interpolation are done to reduce inhomogeneities potentially originating from employing two different generations of the EPS. The objective selection of skilful predictors is done by applying a Stepwise Linear Regression algorithm with Forward Selection and Backward Elimination. Starting with an empty statistical model – during the Forward Selection – all potential predictors are tested whether they provide significant benefit (p<5% according to f-test regarding residual sum of squares) to the model. The best predictor is chosen to enter the model. Subsequently, all remaining predictors are tested equivalently whether they can significantly improve the model. This is done repeatedly until no more predictors yield significant additional value for the statistical model. Finally, the Backward Elimination conversely checks whether predictors can be removed again without significantly decreasing the statistical model's quality (p<10%).

The training of this statistical downscaling procedure and its evaluation (by three-fold cross-validation and several MSE-related metrics) is based on dynamical regionalisation of 181 European winter storm episodes that were done by employing the numerical weather prediction model chain (global model GME and regional model COSMO-EU) of the German Weather Service (DWD).

A comprehensive description of this statistical downscaling approach, as well as its development and evaluation is given by Kruschke (2015). This includes testing various other combinations of potential predictors and demonstrating that this approach





outperforms (measured with respect to the mean squared error of wind gusts) a similar approach described by Haas & Pinto (2012), which is also based on Multiple Linear Regressions. Kruschke (2015) additionally provided an effective quantification of uncertainties of the statistically modelled gusts. However, these uncertainty estimates are not used in the course of the current study.

## 3.2 Ensemble Post Processing

Aim of the ensemble prediction is an estimation of the uncertainty in the prediction. In practice, an ensembles often under- or overestimate the uncertainty, what is meant by the terms  under- and over-dispersion At the ECMWF, the method of singular vectors is used to generate a set of initial conditions which are used to calculate several members of a forecast ensemble with the intention to produce an optimal spread. It should be noted, that the ECMWF EPS has been constructed in the way that its spread is optimised for the medium-range, thus for forecasts of three to five days. Despite of such sophisticated techniques for the perturbations, ensemble forecasts still often tend to an under-dispersion. A post-processing can help to adjust the spread of the ensemble, what is meant in this study by the term « calibration ». Several methods exist to calibrate a forecast ensemble partly depending of the ensemble-type (single-model, multi-model or lagged-averaged-forecasts). An overview of calibration techniques for medium-range forecasts can be found in Gneiting (2014). For this study, we apply the approach of Bröcker and Smith (2008). This method states a so-called ensemble dressing approach, whose purpose is to estimate the probability density function (PDF) of the ensemble, and can be used to adjust the spread. The chosen method has the advantage that it can represent different methods of ensemble dressing depending on the selected parameter set. It transforms the discrete members (50 in our case) to a continuous distribution function, by combing kernel functions for each individual member. The ECMWF EPS is a single model ensemble and all of the members are indistinguishable. For this reason, all members are dressed by using the same Gaussian kernel. The dressing is done using an affine ensemble transformed version of the original data (Bröcker and Smith, 2008). While the dressing is used to transform the discrete members to a distribution function, the affine transformation is used to eliminate biases from the raw forecasts. Parameters for the transformation as well as for the Gaussian kernel are estimated using the minimization of the continuous ranked probability score (CRPS, compare Gneiting and Raftery, 2007). The CRPS is a measure, which describes the performance of an ensemble in its entity by comparing the forecast and observation CDF's.

In general, the aim of the method is the estimation of the entire probability density function (PDF) of forecasts, based on the 50 ensemble members. However, in our case, we are interested in deriving a corrected 50 member forecast ensemble, which is representative for this full PDF. This can simply be accomplished by randomly sampling the 50 members from the calibrated PDF. However, the calibration should not be interpreted for these individual members, since the method is designed to calibrate the ensemble properties (such as ensemble bias and dispersion) rather than the individual members' properties.





### 3.3 Probabilistic Storm Loss Model

In the last step, the forecasts of near surface maximum gusts are translated into probabilistic estimates for the exceedance of specified loss ratio thresholds ("damage occurrences"). Due to insufficient information about meteorological conditions on sub-grid scales (e.g. turbulent gusts induced through localized orographic features) as well as lack of knowledge on individual

building characteristics, it is impossible to model damage occurrences on individual entity level in a deterministic manner. Instead, a statistical relation, valid for the total stock of buildings within a district is derived, which shall enable the specification of probability estimates to express these uncertainties. To do so, logistic regression analysis is performed for each district. Damage occurrences, defined as the exceedance of loss ratio above a certain threshold are derived from the observed loss ratio time series and resulting time series are then related to daily maxima of near surface wind speeds from the COSMO-

EU analyses to train the logistic regression curve. For each district, wind speeds at the closest grid point from the centre of the district are used.

### 3.4 Probabilistic Forecasts of Damage Occurrences

To be able to investigate the influence of the individual uncertainty sources (meteorological forecast uncertainty and damage modelling uncertainty) different probability forecasts are set up. Specifically, four different setups result from i) treating no

uncertainty resulting in deterministic forecasts, ii) treating only meteorological forecast uncertainties, iii) treating only damage modelling uncertainty and iv) a full uncertainty treatment including both uncertainty sources.

The derivation of probability forecasts for damage occurrences is straightforward in the case of individual (single) member forecasts which is done simply by applying the logistic regression function (described in the Sec. 3.3) to calculate a probability estimate for the given forecasted wind speed. Similarly, the logistic regression function can be applied to the ensemble mean.

Resulting probability estimates include damage modelling uncertainty, while neglecting meteorological uncertainties (setup iii). Additionally, meteorological forecast uncertainty information is taken into account by applying the transfer function to each ensemble member. Assuming the members to be equally likely, probability forecasts can then be calculated as the ensemble mean of the damage occurrence probabilities derived for the individual ensemble member forecasts (setup iv). Similarly, to neglecting meteorological forecast uncertainties, the statistical uncertainty from the damage-modelling step can

be neglected by assuming a stepwise function instead of the logistic regression curve (compare Fig. 1, top). This is done by assuming a probability of one in case the forecasted wind speed exceeds a critical threshold and a probability of zero otherwise. Even though not restricted to this choice, we choose this critical threshold to correspond to the wind speed for which the probability from the logistic regression analysis is 0,5. No treatment of uncertainty is accomplished, when applying this "deterministic" damage occurrence function to the ensemble mean forecast (setup i). Finally, probability forecasts can be

generated by applying the "deterministic" damage occurrence function to individual ensemble member forecasts. Probability estimates are then again calculated by averaging over the resulting individual member probability (setup iii). Since in the deterministic case this is either one or zero, this is similar to the fraction of members exceeding the critical wind threshold.





### 3.5 Verification Methodology

The statistically downscaled wind gust ensemble forecasts are investigated on grid-point basis by means of Talagrand diagrams (see e.g. Jolliffe and Stephenson, 2003; Wilks, 2011). A Talagrand (or rank) histogram can be used to illustrate model biases as well as an under- or over- dispersion of the ensemble. To construct the Talagrand diagram, the ensemble members are ordered after their rank for each time step and for each grid cell in ascending order. The frequency of observations falling in between these ranked ensemble members is counted. In a perfect ensemble, each rank would be equally populated, meaning that each ensemble member is equally likely. An asymmetry shows a bias, as too often the ranks of the weakest or the strongest members are populated. If the Talagrand diagram has a u-shape, the ensemble is under-dispersive. This means that the observations are often outside the range spanned by the ensemble. In other words, the ensemble does not cover the entire range of uncertainty. In the opposite case of an over-dispersive ensemble, intermediate ranks of the Talagrand diagram would be overpopulated. This means that observations often lie close to the ensemble median, indicating an overestimation of the involved uncertainty.

Forecast quality of derived daily probability estimates for damages on district level are assessed by means of the Brier Score (Wilks, 2011) which is the mean quadratic error of the probability forecast

$$BS = \frac{1}{N_t}\sum_t(f_t - o_t)^2 \tag{1}$$

where $f_t$ is the forecasted probability at time t and $o_t$ the observation being either one if an event has occurred and zero otherwise. Forecast skill is evaluated with respect to a reference forecast, leading to the Brier skill score

$$BSS = 1 - \frac{BS}{BS_{ref}} \tag{2}$$

with $BS_{ref}$ being the Brier score of the reference forecast. In the course of this study we use the climatological forecast, i.e. the climatological event frequency as reference. Separately for each district, Brier Scores and Brier Skill Scores are evaluated. To summarize the verification statistics, Brier score and Brier Skill Score are evaluated additionally based on all forecast times and all $N_d$ districts.

$$BS_{tot} = \frac{1}{N_d}\frac{1}{N_t}\sum_d\sum_t(f_{t,d} - o_{t,d})^2 \tag{3}$$

where $f_{t,d}$ is the forecasted probability at time t in district d and $o_{t,d}$ the corresponding observation. It should be noted that districts are equally weighted in Eqn. 3, disregarding differences in size. It might be argued, that this leads to an overweighting of small districts, e.g. urban municipalities. However, in our study we omitted such weighting since typically the sum of insured values is higher in these urban municipalities, justifying such higher weighting.

Confidence intervals on derived Brier Scores are calculated by means of a bootstrap method, randomly generating 10000 $BS_{tot}$ values. This sampling is accomplished by randomly drawing $N_d \cdot N_t$ times from the original set of individual contributions $(f_{t,d} - o_{t,d})^2$ to the total Brier score in Equation 3. Confidence intervals on $BS_{tot}$ are then calculated as the 5% and 95% quantiles of the 10000 randomly generated $BS_{tot}$ values. Differences in the Brier Skill Score are considered significant, if the derived 90% confidence intervals are exceeded.



## 4 Results

### 4.1 Verification of severe wind gust predictions based on statistically downscaled EPS

In a first step, the statistically downscaled ensemble forecasts were verified against the COSMO analyses, by means of the rank histogram statistics described in Sec. 3.5. The resulting Talagrand diagrams for forecast lead times of 1, 3 and 9 days

(red, green and blue, respectively) are shown in Fig. 2 (left). The first thing to note is that there is an asymmetry to the right hand side. For 1 day forecast lead time it is found that in nearly 40% of the cases, the observation is equal or above the largest value of the ensemble. At first sight, such frequency bias appears to be rather critical. However, the absolute bias of the downscaled ensemble forecasts (not shown) ranges only in between 0.1-0.5m/s, depending on the grid box considered. Furthermore, considering the conditional bias of the ensemble forecasts (not shown) revealed that this underestimation occurs

mainly for low forecasted gusts. Still, the application of this dataset for storm damage modelling would lead to an underestimation of the estimated storm damage probabilities. The second thing to note in Fig. 2 (left) is the underdispersion, demonstrated by the u-shape. As described in Sec. 3.5, this indicates an underestimation of the uncertainty on forecasted wind gusts. With increasing forecast lead time, both u-shape as well as the asymmetry in the Talagrand diagram decreases (Fig. 2, left). This means that both underdispersion and frequency bias decrease with increasing forecast lead time, which might relate

to the fact that the ECMWF EPS system as mentioned above is primarily designed for forecasts in the medium range. Thus, the ensemble spread is assumed to be optimised for lead times of several days, for shorter lead times however this might not be the case.

To correct both bias and underdispersion, the ensemble post processing technique after Bröcker et al. (2008) was applied to the data. Considering the Talagrand diagrams for the post processed forecast (Fig. 2, right) shows nearly equally populated

ranks. Slightly higher populations (doubled in case of forecast lead time of 1 day) are found for the lowest and highest ranks. In only 4% of all forecasts, the observation falls below the lowest value and above the highest of the ensemble forecast members. Thus, the underdispersion is largely removed by the post processing. For increasing lead time the remaining underdispersion further declines. Also the Talagrand histograms for the post processed ensemble (Fig. 2, right) show no considerable asymmetry, indicating that the bias found for the downscaled forecasts are removed.

### 4.2 Prediction skill of storm loss occurrences

The four different settings (as described in Sec. 3.4) are used to forecast storm damage occurrences from the statistically downscaled EPS forecasts. As an illustrative example, resulting forecasts on district level are visualized in Fig. 3 for October 31st 2006 (winter storm Britta). In about half of all 439 districts, the loss ratio within individual districts exceeded the threshold 0.0001‰. For a lead time of 1d (forecasts initialized on 12 UTC of the previous day) the deterministic setup (no uncertainty

treatment) forecasts such exceedance in considerably less districts. With a treatment of meteorological uncertainty, non-zero probabilities for damage threshold exceedances are derived in a number of districts, for which the deterministic model does not forecast a threshold exceedance. However, still large areas which had been affected by damages feature only probabilities



below 10%. The treatment of damage occurrence uncertainty in case of Storm Britta yields a rather different picture. Now, probabilities of 20% or higher are derived for most northern regions which featured damages. Particularly considering the dressed ensemble forecasts, forecasts applying a full uncertainty treatment feature probabilities higher than 40% on most regions affected while probabilities of 10-20% in the southern regions in which only few individual districts featured damages.

Considering longer lead times, it shows that using the full uncertainty information (particularly by means of the dressed ensemble) seems to be advantageous compared to the methods disregarding uncertainty information. In this example, using full uncertainty information even 9 days in advance yields probabilities of 10-20% in most of the areas affected while neglecting the uncertainty information does not yield any signal with respect to damage occurrences.

Of course, the quality of probabilistic forecast can't be judged by means of single forecasts or single storm situations. Instead,
a systematic evaluation of forecast quality is performed by means of objective verification. Verification of damage occurrence forecasts is performed for exceedances of a lower threshold (loss ratio > 0.0001‰) as well as a higher threshold (loss ratio > 0.001‰). Climatological occurrence frequencies for events defined in this way range from 9 to 45 days per winter half year for the former, and 1 to 11 days per winter half year for the latter depending on the district considered. In average over all districts, climatological event frequencies are about 20.9 days per winter half year (11.5% of days) for the low threshold and
3.5 days (2%) for the high threshold. It should be noted, that the events exceeding the higher threshold are a subset of the events exceeding the lower threshold. However for readability, we call the former set of events "low impact events", since in terms of the occurrence frequency the lower impact events strongly dominate (by a factor of about 6).

Considering the Brier Skill Score (as described in Sec. 3.5) with the climatology as reference forecast it is confirmed, that the
deterministic forecasts of damage occurrences only yield very low skill on the first forecast day (compare circles in Fig. 4). Considering meteorological uncertainties for low impact events (loss ratio > 0.0001‰), significant forecast skill is achieved for up to 6 days lead time (Fig. 4, left). However, skill is strongly increased if the damage model uncertainty, namely the statistical uncertainties within the relation between wind and damage occurrence probability, are treated. For lead time of 1 day the BSS raises from about 0.1 to nearly 0.3. Treating the damage model uncertainty yields skilful forecasts for the whole
range of lead times considered. For low impact events it shows that an explicit treatment of both uncertainties only yields small additional value, indicating that uncertainty in this case is dominated by the damage model uncertainty. Only for long lead times, for which meteorological forecast uncertainties naturally grow, an additional advantage is generated by the explicit and full treatment of both uncertainty sources. For lead times of 9 days this advance in forecast skill corresponds to a gain of about 1 day in lead time.

The situation is different in case of high impact events. Even for lead time of 1 day, treating both uncertainty sources yields a significant advantage compared to the other methods. This can be understood by considering that for the high impact situations (featuring severe wind conditions) even on such short lead times considerable meteorological uncertainty (on forecasted wind speeds) is present. The gain of forecast skill (by means of full uncertainty treatment) again increases with increased lead time. This of course is due to growing meteorological forecast uncertainty which is even larger in case of the high impact events.





Compared to the methods treating only individual uncertainty sources, the full treatment of uncertainty leads to an improvement of forecast skill corresponding to about 2-3 days for lead times up to 9 days.

Additionally, it is found that the ensemble post processing method (as described in Sec. 3.2) leads to an improvement in forecast skill. This improvement is found to be stronger and statistically significant for short lead times and particularly in the
case of high impact events. This is consistent to the finding that both bias as well as underdispersion are stronger at short lead times. Particularly in case of high impact events the correction of bias and underdispersion results in a gain in forecast skill corresponding to about 1-2 days lead time.

The spatial stratification by districts shows that forecast skill is not homogeneous over German districts (Fig. 5). In general, higher skill is found in northern regions. It can be assumed that this higher skill in northern regions is due to an increasingly
flat orography. Over complex terrain, predictability of wind gusts can generally be assumed to be lower, which is thus consistent to the spatial differences with respect to the predictability of damage occurrences.

Furthermore, the spatial stratification also shows, that skilful forecasts throughout Germany are only achieved through a treatment of the damage model uncertainty (Fig. 5), even for the shortest lead time of 1 day. Further improvement is achieved by full treatment of uncertainty, which has been quantified in the previous paragraphs.

**5 Conclusions and Discussion**

A probabilistic approach to forecast local occurrences of damages due to severe winds was presented. The approach is based on a logistic regression analysis, relating daily maxima of near surface wind speeds from meteorological analysis data to damage occurrences for individual districts within Germany, defined through the exceedance of the loss ratio over a specified threshold. Due to unknown meteorological conditions on subgrid scales as well as unknown details on individual housing
characteristics, it is impossible to model damage occurrences on individual building level in a deterministic manner. Instead only a statistical relation valid for a certain stock of buildings within a district can be derived. The probability estimates for specific wind speeds then reflect the "damage occurrence uncertainty" arising from unknown details on unresolved spatial scales.

Forecasting winter storm damages, further uncertainty arises due to meteorological forecast uncertainties. In this study, these
uncertainties were addressed by applying the storm damage model to the operational EPS system of the ECMWF. Since the resolution of the ECMWF EPS is too coarse, a statistical downscaling was applied to obtain near surface wind gusts on the COSMO-EU grid (7km).

In a first step, the statistically downscaled wind speeds were verified against analyses, indicating a bias of the ensemble predictions towards lower wind speeds. In addition, the ensemble predictions were found to be underdispersive, thus showing
too few ensemble spread, which indicates an underestimation of uncertainty by the ensemble.

By applying the probabilistic storm damage model to the ensemble forecasts the influence of the individual uncertainty sources (meteorological forecast uncertainty and damage model uncertainty) has been investigated.



Results show that neglecting the statistical uncertainty arising within the damage model leads to rather poor forecast skill. Particularly for low impact events and for small lead times, the damage model uncertainty is found to dominate the overall uncertainty. This reflects the fact, that meteorological forecast uncertainties are obviously small at short lead times and particularly in the case of low impact (low wind) situations where basically an ensemble mean forecast or even a single

deterministic forecast is sufficient to derive reasonable forecasts.

With longer lead times meteorological forecast uncertainties naturally play an increasing role. Particularly for high impact situation (due to severe winds) it was shown that meteorological forecast uncertainties cannot be neglected without severe deficiency in skill. This means that an explicit treatment of both uncertainties leads to strong improvement of forecast skill.

The reason for this can be found in the nonlinearity of the relation between the meteorological parameter wind and resulting

impact or impact probability. Basically such nonlinear relation implies the necessity of weighing ensemble members in a more complex fashion compared to calculating simply the ensemble mean wind speeds. This nonlinear weighing is taken into account by the impact modelling step and subsequent ensemble averaging for the forecast quantity of interest (in this case impact probability). Thus in such situation an explicit treatment of uncertainty through the complete modelling chain is highly beneficial.

As stated, for short lead times and low impact situations this effect due to an explicit and full uncertainty treatment is negligible. For large lead time (up to 9 days) this effect corresponds to a gain in forecast lead time of one day. For high impact situations this effect is even stronger corresponding to a gain of 2-3 days lead time.

Both bias and underdispersion of the ensemble forecasted wind speeds have been treated by applying an ensemble post processing method (ensemble dressing) which is found to effectively compensate both shortcomings. Using the ensemble

dressed wind speeds as the basis for the damage occurrence forecasts shows additional forecast skill corresponding to a gain of 1-2 days lead time. This gain is particularly strong at shorter lead times of a few days, for which a stronger bias as well as a stronger underdispersion in forecasted wind speeds has been found.

Overall this study shows, that skilful predictions of storm loss occurrences on lead times of several days can be made using the presented (fully probabilistic) framework to integrate meteorological forecast uncertainties and uncertainties resulting from

impact model. Findings thus demonstrate their potential use in risk based warning systems.

**Acknowledgement**

This research was carried out in the Hans-Ertel-Centre for Weather Research. This research network of Universities, Research Institutes and the Deutscher Wetterdienst is funded by the BMVI (Federal Ministry of Transport and Digital Infrastructures). Furthermore, contributions to this work have been funded by the Federal Ministry of Education and Research in Germany

(BMBF) through the research program MiKlip (FKZ: 01LP1104A) and by Munich Re. We wish to thank the Gesamtverband der Deutscher Versicherungswirtschaft e.V. (GDV) for providing the loss data. We are also grateful to the German Weather Service (DWD) and the ECMWF for providing access to the EPS data.





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



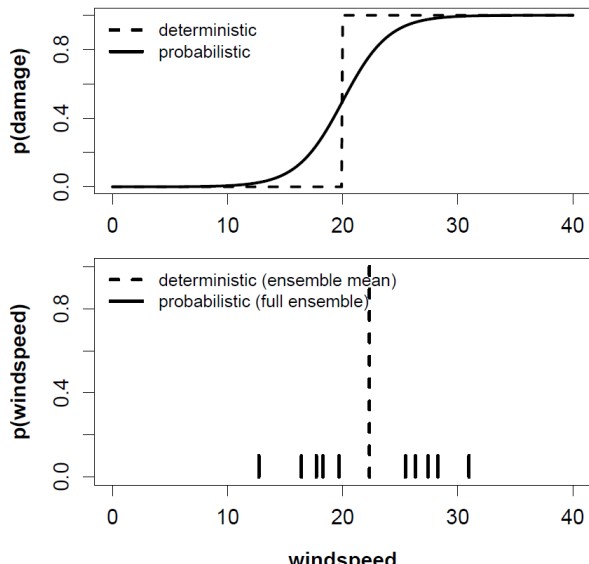

**Figure 1: Illustration of the methodology to derive probabilistic impact prediction from ensemble forecasted wind speed. (top) Probabilistic storm damage function – logistic regression curve – relating the forecasted wind speed to a probability of damage occurrence. The dashed line indicates the deterministic version of such damage function being zero below the critical wind threshold and one above, respectively. (bottom) Illustration of wind speeds forecasted by a 10 member ensemble in solid lines. Dashed line indicates the ensemble mean.**





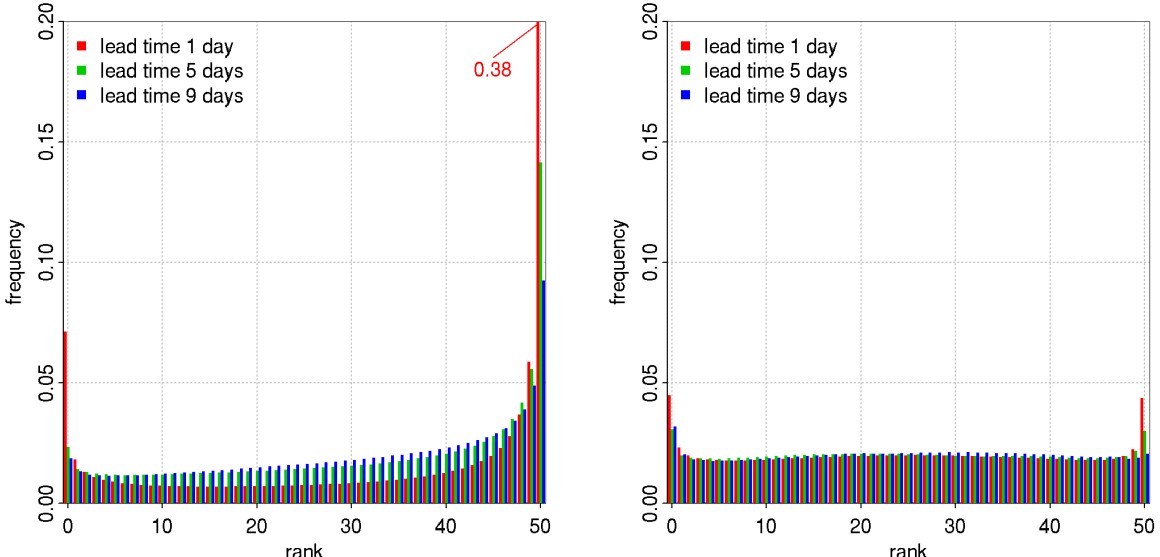

**Figure 2: (left) Talagrand diagram of statistically downscaled EPS forecasts, lead time 1 day (red), 5 days (green) and 9 days (blue) for January 2006 to January 2010. (right) Talagrand diagram of statistically downscaled and post processed EPS forecasts, lead times 1, 5 and 9, January 2006 to January 2010.**




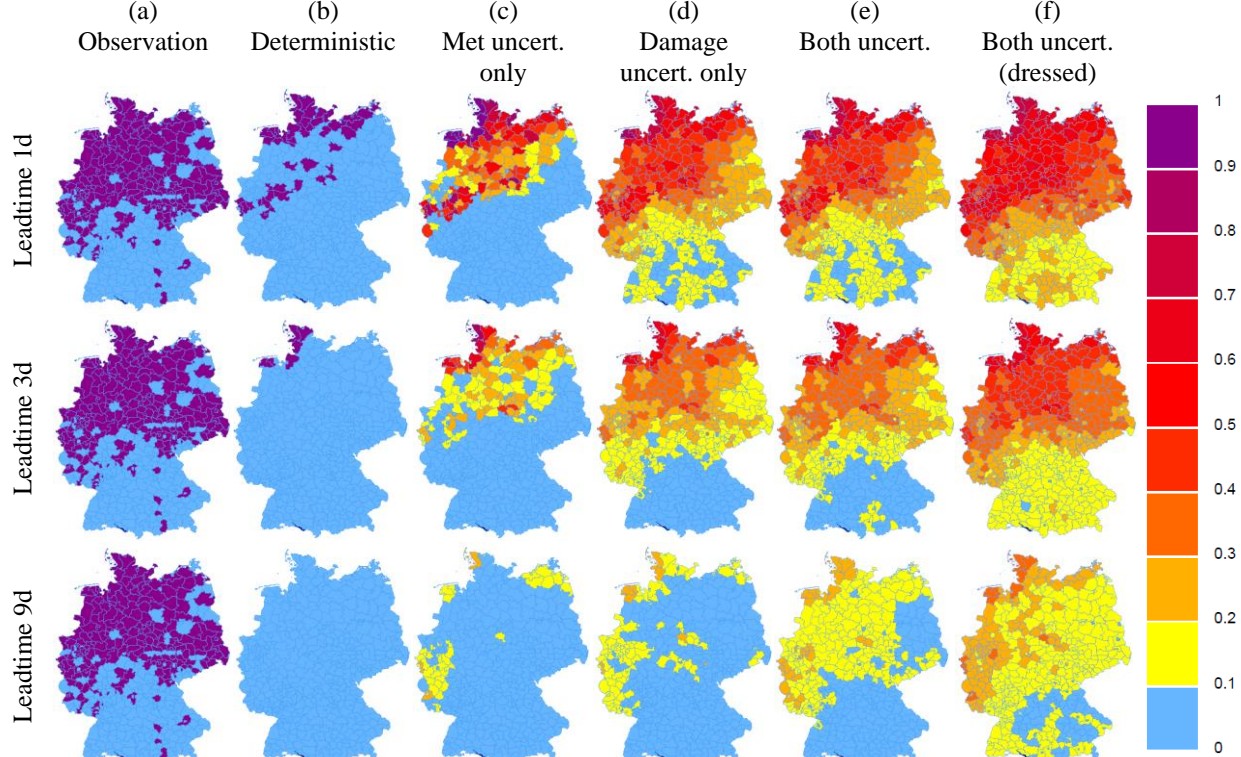

Figure 3: Observed occurrences and forecasted probabilities for loss ratios exceeding 0.0001‰ for October 31st 2006 (Winter storm Britta). (a) Observation. (b) Deterministic forecast disregarding both uncertainty sources. (c) Only considering the meteorological uncertainty. (d) Only considering the damage modelling uncertainty. (e) Considering both uncertainties. (f) Considering both uncertainties, based on the dressed ensemble. (© GeoBasis-DE / BKG 2008)




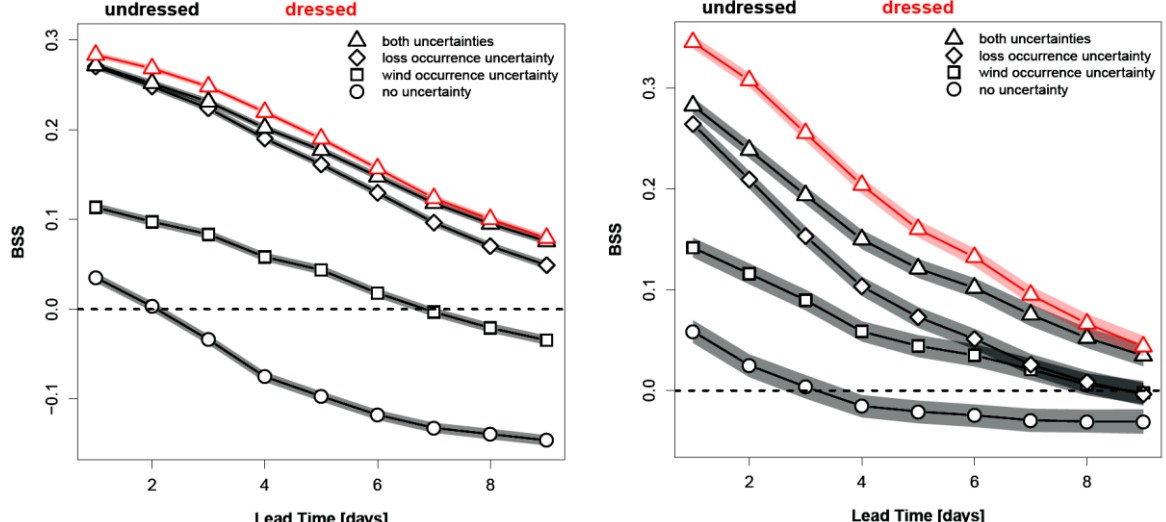

**Figure 4: Lead time dependent Brier Skill Score (BSS; employing climatology as reference forecast) for events with loss ratio exceeding 0.0001‰ (left) and loss events with loss ratio exceeding 0.001‰ (right). Shown in black symbols are verification results for the 4 different setups, red triangles shows verification results using the post processed ensemble. 90% confidence intervals from a bootstrapping method are shown as shaded areas.**



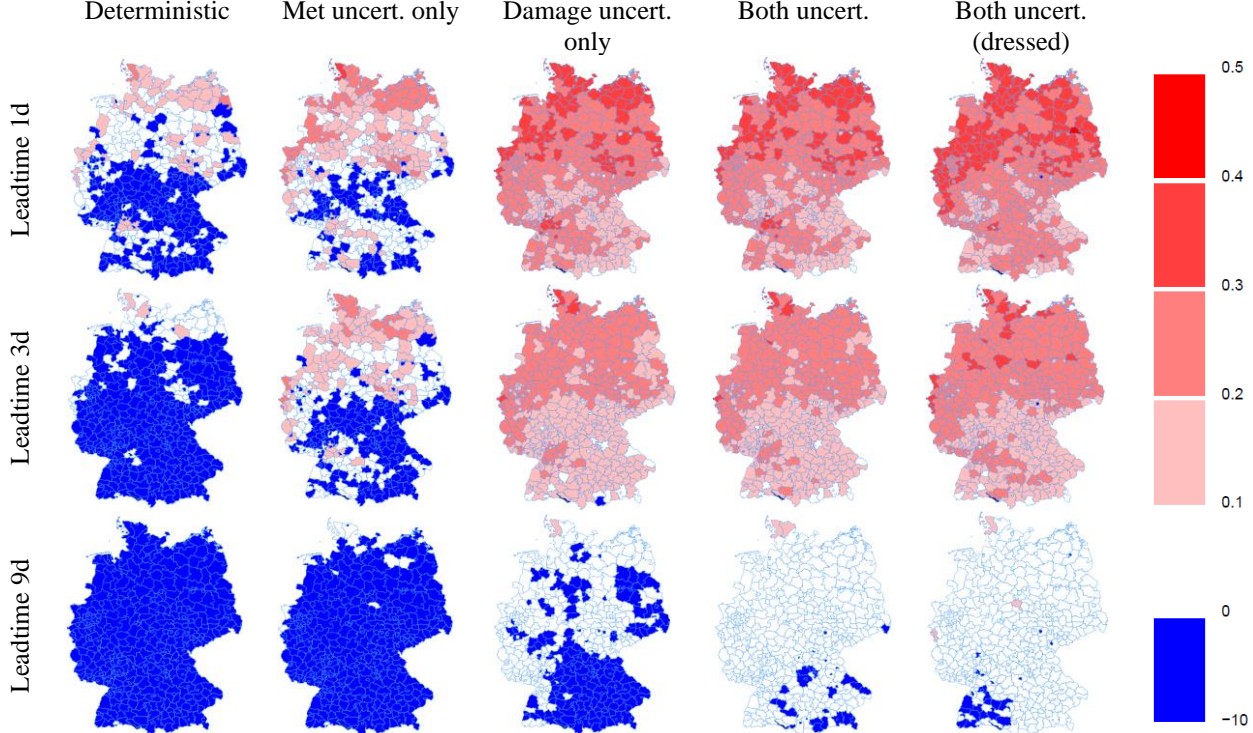

**Figure 5: Brier Skill Score (BSS; employing climatology as reference forecast), Loss threshold 0.0001‰. (©
GeoBasis-DE / BKG 2008)**