# Peer review of "Uncertainties in Forecasts of Winter Storm Losses"

_Natural Hazards and Earth System Sciences, 2016_

## Referee Comment (RC1) · Anonymous Referee #1 · 11 Aug 2016

Overall quality of the discussion paper ("general comments"): This study focuses on the meteorological and damage uncertainties and how quantification of these can derive probabilistic forecasts of winter storm damage over Germany. Model forecast uncertainties are verified with and compared to observation data on insurance losses. Statistical techniques are used to estimate damage uncertainties with appropriate verification methodologies being used to analyse forecast skill.

Forecasting losses from winter storms is extremely difficult, this paper makes good strides in how to address these difficulties in a novel way and provides a good methodology for quantifying uncertainties in the factors which make loss forecasts so complex. The insurance data provides a unique dataset which is rarely seen in these kinds of studies.

[Figure]

Sections 2 and 3 (Data and Methodology) have good content and explanations. Assumptions are outlined well and scientifically backed with appropriate evidence. The investigation analysing the influence in different uncertainty types, four in total plus one dressed ensemble post-processing approach) and their interactions is well thought out and a good method for this kind of analysis.

Overall an interesting paper which should be published after some minor corrections (see below). Clarification is required in some areas to make the paper more easily understandable along with numerous technical corrections.

Individual scientific question/issues ("specific comments")

Title: Could better reflect the study for example 'An Analysis of Uncertainties in Forecasts of Winter Storm Losses'.

Page 1 line 12 (abstract): 'Deterministic assessment of damages' please clarify, is this prior to the event? Even knowing local vulnerabilities would not allow for a yes/no damage assessment until after the event when the damage is done.

Page 3 lines 8-11: It is stated that 'hail induced damaged cannot be separated...this poses another uncertainty that needs to be reflected in the relationship between local winds and resulting damages' however this is not mention again in the paper. How this could be done and what it may show?

Page 5 line 12: How can post-processing help adjust the spread of an ensemble? Can this be made more explicit/clearer?

Page 5 line 20: Can the size of the Gaussian kernel be specified?

Page 8 line 10: Clarify what 'low forecasted gusts' are.

Page 8 line 20: Clarify this sentence, how is a doubling shown? The sentence and Figure don't appear to correlate.

Results section: Over-forecasting is not mentioned. On page 9 lines 8 and 9 '10-20%

probability using full uncertainty in most of the affected areas' is stated. I understand the nature of probabilities however it would be good to have a comparison between forecast probabilities for the loss ratios and in the areas where damage was observed i.e. how often is 10-20% forecast and damaged observed?

Page 9 line 10: What is meant by 'objective verification'?

Page 10 lines 8-14: Could anything else be said about Fig 5? Perhaps correlate BSS to orography? Comment why the BSS in some areas is so low at -10? Could Fig 5 be related to the forecasts in Fig 3?

Page 10 line 5: Don't agree with the last part of this last sentence, it depends on the meteorological situation.

Page 10 line 15: Clarify 'this effect'

Page 10 line 25: Could 'how these findings could potentially be used in risk based warning systems' be expanded? More explicitly how could forecasters use this approach to improve forecasts and warnings?

Technical corrections

Generally:

Define when talking about mean wind speeds and wind gust speeds. For wind impacts and studies using wind data knowing the difference is important.

Ensure paragraphs aren't one sentence long.

Keep consistent phrasing e.g. 'damage uncertainty' not 'impact model uncertainty' (e.g. page 2 line 14); 6-hourly (page 4 line 1) and 6 hourly (page 3 line 16).

Be consistent in the use of 'damage modelling uncertainty', 'meteorological uncertainty' and 'full uncertainty/both uncertainties' e.g. Page 9 line 1 it's referred to as 'damage occurrence uncertainty' and page 9 line 5 vs Figure 3 and Figure 4.

Ensure consistency of Brier Score and Brier Skill Score capitalisation or define then use BS and BSS don't mix.

Consistency needed for 'winter storm Britta' references including removing capitalisations.

Change all references of 1d, 3d and 9d to 1 day, 3 days and 9 days respectively including in figures.

Keep 'low' and 'high' threshold wording consistent, 'lower' and 'higher' thresholds are referred to on page 9.

Check references in particular comma placements and journal titles.

Page 1 line 14: clarify what are the 'two individual contributions'

Page 1 line 15: 'forecast skill' of what?

Page 1 line 17: define 'district level'

Page 2 line 8: source 'of' uncertainty not 'for'

Page 2 line 17: approach 'when' predicting

Page 2 line 26: comprise 'of' daily data

Page 3 line 2: 'insurers' not 'insurances'?

Page 3 line 15: define DWD

Page 3 line 16: 10 meter's' above 'the' ground

Page 3 line 18/19: 'hourly 10m wind gusts' instead of 'hourly wind gusts in 10m height'

Page 3 line 20: 'maximum 10m wind gusts' instead of 'maxima of wind gusts in 10m height'

Page 3 line 22: ECMWF 'has' operationally 'run' its

Page 3 line 24: can remove 'the'

Page 3 line 25: check sentence structure and TL use

Page 3 line 28: 'In' December or re-arrange sentence

Page 4 line 1: remove space between 10 and m

Page 4 line 11: 436,905 not 436.905; comma needed after 'equation'

Page 4 line 15: TL not TL

Page 4 line 18: maximum 10m 'wind' gusts

Page 5 lines 6 and 7: Makes very little sense, please revise

Page 5 line 11: 'Post-processing' no 'A' needed.

Page 5 line 12: 'what is meant in the study by the term « calibration »'? please revise

Page 5 line 13: depending 'on' the, not 'of'

Page 5 line 25: What are CDF's?

Page 5 line 26: PDF already defined

Page 6 line 6: move comma to after district and remove from after derived

Page 6 line 18: no 'the' needed before Sec. 3.3

Page 8 line 8: 'range is only between'

Page 8 line 15: 'previously' not 'above'

Page 8 line 20: Don't need 'Considering' at the beginning of the sentence.

Page 8 line 28: add 'observed' to make 'the observed loss ratio'

Page 8 line 30: add 'only' after 'uncertainty' and clarify/reword the sentence.

Page 9 line 1: 'uncertainty only for winter storm Britta' or 'uncertainty in the case of

winter storm Britta'

Page 9 lines 2-4: reorder to refer to both uncertainties then dressed ensembles, this fits better with the order of Figure 3.

Page 9 line 9: 'forecasts' not 'forecast'

Page 9 line 13: 'When averaged over' not 'In average over'

Page 9 line 19: '3.5), with the climatology as a reference forecast, it is confirmed that'

Page 9 line 23: 'For a lead time'

Page 9 line 30: define 'high impact events'

Page 9 line 30: 'Even for a lead time'

Page 10 lines 4 and 5 and Page 11 lines 8 and 17: not sure 'stronger' and 'strong' are the correct words, please revise

Page 10 line 6: 'in the case'

Page 10 line 16: 'winter storms' instead of 'severe winds'

Page 10 line 24: 'In forecasting'

Page 10 line 30: 'too little ensemble'

Page 10 line 2: 'for short lead times'

Page 10 line 3: no comma needed after 'fact'; 'uncertainties are smaller at' no need for 'obviously'.

Page 10 line 7: 'situations'

Page 10 line 8: 'improvement in forecast'

Page 10 line 16: 'large lead times'

Page 10 line 23: 'Overall, this study shows, for this particular case study, that'

Figure 3: Units on right hand side, referred to in % in text but decimals in figure.

Figure 4: Make scales the same so easier to compare. Caption could say ' Lead time dependent Brier Skill Score (BSS; employing climatology as the reference forecast) for events with loss ratio exceeding low threshold (0.0001o/oo) (left) and loss events with loss ratio exceeding high threshold (0.001o/oo) (right). Shown in black symbols are verification results for the four different set-ups, red triangles show verification results using the ensemble dressing post processing system.'

Figure 5: 'for loss ratio exceeding low threshold (0.0001o/oo)'

---

## Referee Comment (RC2) · Anonymous Referee #2 · 1 Sep 2016

The authors investigate medium-range predictions of winter storm losses in Germany, taking into account uncertainties related to both the meteorological forecast and the storm loss model. While the deterministic forecasts had little or no skill for lead times beyond one day, the authors show that the prediction skill can be considerably improved when taking the two types of uncertainty into account in a probabilistic framework. The skill improvements are highest when both uncertainties are taken into account, enabling skilful loss predictions several days ahead.

General comments:

The paper is clearly written and logically organised, it reads very well and the methodologies are well explained. Apart from one more general comment that can hopefully be addressed easily, I mainly found a couple of wording issues or typos that need to

be corrected. After some minor revisions the paper should be suitable for publication.

On the example of one storm (pages 8/9) the authors demonstrate how accounting for the uncertainties increases the probability of losses occurring, and this improves the forecast for the case when a storm / damage occurs as the deterministic model strongly underpredicted the losses. However, I am wondering if this also increases (and by how much) the probability of damage when no storm occurred, thereby leading to a higher false-alarm rate? I would appreciate if the authors could expand on issues around false alarm rate.

Specific comments:

Title: the paper does not only report on uncertainties but also quantifies skill and shows how treating uncertainties increases the skill. Therefore I think it would be good to add "skill" to the title, e.g. "Uncertainties and skill in forecasts. . ." or similar

Section 2: As the different data (loss, COSMO, ECMWF EPS) cover different time periods, it would be good to say explicitly for which period the skill calculations were performed, for which period the downscaling was trained, etc.

Technical corrections:

page 1, line 30: not all references seem to be in the reference list at the end of the paper

page 2, line 27: would "records" be more appropriate then "measurements" when talking about the insurance data?

page 3, line 4: typo: inhomogeneities

page 5, line 7: end of sentence behind "over-dispersion"

page 5, line 12: comma at end of line behind "forecast ensemble"?

page 5, line 28: representative of

page 6, line 9: split the long sentence behind "loss ratio time series. The resulting. . ."

page 6, lines 19/26: past tense of forecast is forecast

page 6 section 3.4, it seems there is no explicit reference to setup ii)

page 6, line 28: 0.5 (decimal point rather than comma)

page 7, line 5: "ordered according to" rather than "after"

page 7, line 16: should the "and" (after has occurred) better be "or"?

page 9, lines 2/4: consider replacing "featured" by "recorded"

page 9, line 21: do you show "significant" or rather "positive" forecast skill?

page 10, line 5: consistent with

page 10, line 9: may the higher skill in northern regions also be due to higher average loss ratios in these districts?

Section 5: most of this sections reads more like a "Summary" rather than "Conclusions and Discussion". I figure that the very last 3 lines (page 11, lines 23-25) may be your main conclusion?

page 10, line 28: specify "meteorological analyses"

page 10, line 30: too little ensemble spread

page 11, line 3: remove comma behind "fact"

---

## Author Comment (AC1) · 6 Oct 2016

**Reply to RC1**

The authors would like to thank the reviewer for the very thoughtful review. The comments, suggestions and numerous technical corrections will certainly help us improve the manuscript. Point-by-point answers (in *italics*) to the comments are given in the following.

**Individual scientific question/issues ("specific comments")**

Title: Could better reflect the study for example 'An Analysis of Uncertainties in Forecasts of Winter Storm Losses'.

*New title based on both reviewers' comments: "An Analysis of Uncertainties and Skill*

[Figure]

*in Forecasts of Winter Storm Losses"*

Page 1 line 12 (abstract): 'Deterministic assessment of damages' please clarify, is this prior to the event? Even knowing local vulnerabilities would not allow for a yes/no damage assessment until after the event when the damage is done.

*We exchanged "..deterministic assessment of damages.." by "..deterministic prediction of damages, even if the forecasted wind speed contains no uncertainty"*

Page 3 lines 8-11: It is stated that 'hail induced damaged cannot be separated...this poses another uncertainty that needs to be reflected in the relationship between local winds and resulting damages' however this is not mention again in the paper. How this could be done and what it may show?

*This is a very interesting and tricky point! From a data point of view, hail induced damages cannot be distinguished from wind related damages in the dataset we use. According to the provider of the dataset (GDV), winter months are dominated by windstorm damages while summer is dominated by hail induced damages. However in rare cases of severe winter storm events, hail damages may occur. E.g. it is known that damaging hail occurred during the frontal passage of storm Kyrill in 2007 (Fink et al, 2009). Taking into account the occurrence of hail and resulting damages could be done based on additional predictors such as the "convective available potential energy" (CAPE) and "convective inhibition" (CIN). Based on a logistic regression model with multiple predictors, both the individual effect of hail but also the contribution of hail in the case of winter storms could be quantified. It can be assumed, that the probability of hail will increase in case of the most severe winter storm events. Thus for high wind speeds, the damage probability forecasts (which neglect the effects of hail) might be underestimated. Considering the reliability diagrams for the probabilistic forecasts (not shown) we do find such underestimation of the probability forecasts. However a more in depth analysis is needed to clearly attribute this to effects due to hail. This has not been the scope of this paper but we plan to address this in further research.*

*Furthermore we will add the discussion outlined here to the discussion section within the manuscript.*

Page 5 line 12: How can post-processing help adjust the spread of an ensemble? Can this be made more explicit/clearer?

*We added clarification by modifying the original quote (Page 5, lines 10-12):*

*"Despite of such sophisticated techniques for the perturbations, ensemble forecasts still often tend to an under-dispersion. A post-processing can help to adjust the spread of the ensemble, what is meant in this study by the term 'calibration'."*

*To:*

*"Despite of such sophisticated techniques for the perturbations, ensemble forecasts still often tend to be under-dispersive. This means, that the spread of the ensemble members (the members being discrete random draws of the forecast PDF) may be too small and it may not reflect the full uncertainty inherent to the forecast. 'Calibrating' the ensemble spread, which is part of sophisticated post-processing techniques can thus help address such under-dispersion of ensemble forecasts (see Broecker et al.,2008)."*

Page 5 line 20: Can the size of the Gaussian kernel be specified?

*The ensemble post-processing is performed on a grid-cell basis and the Gaussian kernel determined individually for each grid-cell. Additionally they are dependent on the actual forecast. The kernel size can thus strongly vary depending on grid point and specific forecast situation. Further details can be found in Broecker, J. Smith, L, 2008. For clarification, we will comment this in the manuscript.*

Page 8 line 10: Clarify what 'low forecasted gusts' are.

*To clarify, we revised the sentence to:*

*".. is stronger for weak gusts (<5m/s)"*

*instead of:*

*"..occurs mainly for low forecasted gusts".*

Page 8 line 20: Clarify this sentence, how is a doubling shown? The sentence and Figure don't appear to correlate.

*The original quote read:*

*"Considering the Talagrand diagrams for the post processed forecast (Fig. 2, right) shows nearly equally populated ranks. Slightly higher populations (doubled in case of forecast lead time of 1 day) are found for the lowest and highest ranks."*

*Figure 2 (right) shows a rather flat Talagrand diagram. For most ranks a frequency of slightly less than 0.025 are found. Considering forecasts of lead time 1 (red), the ranks 0 and 50 are populated with a frequency of about 0.05 which is about doubled compared to the other ranks. For longer lead time the overpopulation (underdispersion of the ensemble) is smaller which is why we generalized the statement that for the lowest and highest ranks slightly higher populations are found. We thus think that this statement does correlate to the presented Figure. However, to clarify we revised the above written quote to:*

*"Considering the Talagrand diagrams for the post processed forecast (Fig. 2, right) shows nearly equally populated ranks. Slightly higher populations are found for the lowest and highest ranks. In case of forecast lead time of one day (red), the lowest and highest rank are populated with a frequency of about 0.05 which is roughly twice the frequency found for the intermediate ranks."*

Results section: Over-forecasting is not mentioned. On page 9 lines 8 and 9 '10-20% probability using full uncertainty in most of the affected areas' is stated. I understand the nature of probabilities however it would be good to have a comparison between forecast probabilities for the loss ratios and in the areas where damage was observed i.e. how often is 10-20% forecast and damaged observed?

*This is a very good suggestion. We will add a discussion on the reliability of the probabilistic forecasts. For this, we will add a figure [see additional Fig. 1] to the manuscript containing exemplarily the reliability diagrams for the forecasts with lead time 3 days. Compare Wilks (2008), Chapter 8 for details on reliability diagrams.*

*In case of the deterministic forecasts (black circles) they show, that in about 3% of all cases for which the forecasts reads "no event" a loss event has actually been observed. Similarly in about 97% of the cases for which an event is forecasted a loss event actually occurred.*

*Considering the probabilistic forecasts we find, that if forecasted probabilities are low (<5%), very few events are observed. Using the forecasts treating both uncertainties and using the dressed ensemble results in an observed relative frequency of only 0.4% (which is a considerable improvement compared to 3% in case of the deterministic forecast "no event").*

*The diagrams show that in general an under-forecasting occurs for the probabilistic forecasts. Over a broad range of probabilities the observed relative frequency of events is found to be considerably higher than forecasted. As an example, one might consider all cases in which the forecast probability reads 30%. According to the reliability diagram, in 40-60% of these cases (depending on which uncertainties are treated) an event had been observed.*

*The diagrams furthermore show that this under-forecasting is successively reduced (and thus the reliability increased) by explicitly treating the different uncertainty sources. Particularly for intermediate forecast probabilities, the distance of the reliability diagram to the diagonal (representing the optimum: forecast probability = observed relative frequency) is reduced and is lowest if both uncertainty sources are treated. Also, it can clearly be found, that the reliability is further increased when using the dressed ensemble instead of the raw ensemble forecasts.*

Page 9 line 10: What is meant by 'objective verification'?

[Figure]

*To be more specific, we revised the statement*

*"by means of objective verification"*

*to*

*"by means of Brier Score and Brier Skill Score, which are objective measures for the quality of probabilistic forecasts."*

Page 10 lines 8-14: Could anything else be said about Fig 5? Perhaps correlate BSS to orography? Comment why the BSS in some areas is so low at -10? Could Fig 5 be related to the forecasts in Fig 3?

*Our results show distinct variations over Germany. Even though we did not calculate this, we know that there is a positive correlation. However, technically there are several further questions when correlating the BSS results (on district level) for example with a gridded orography which itself varies across the district. Thus we do not think there is further gain of information by correlating BSS and orography. The BSS is calculated as 1-BS/BSref (compare eqn. 2). Consequently, the BSS potentially ranges from 1 (perfect forecast) to minus Infinity (perfect reference). Since BSref in our study is rather small in most cases, negative BSS tends to quickly grow very large. The relevant information in this case is actually that the BSS is negative (i.e. a worse than the climatology). The scale is thus chosen from -10 to 0 to ensure that all negative BSS values are colored in blue. This does not mean that BSS values are -1 though. We will simply remove the -10 from the scale to avoid this confusion!*

Page 10 line 5: Don't agree with the last part of this last sentence, it depends on the meteorological situation.

*The reviewer is right, the underdispersion is different depending on the meteorological situation. On average we find a stronger underdispersion at short lead times, however we will modify our statement reflecting the reviewers point.*

Page 10 line 15: Clarify 'this effect' (this refers to Page 11 line 15)

*We agree that the sentence needs some clarification. We rewrite the sentence "As stated, for short lead times and low impact situations this effect due to an explicit and full uncertainty treatment is negligible."*

*to*

*"As stated, for short lead times and low impact situations the effect from a full uncertainty treatment is negligible."*

Page 10 line 25: Could 'how these findings could potentially be used in risk based warning systems' be expanded? More explicitly how could forecasters use this approach to improve forecasts and warnings?

*The original quote (Page 10, lines 13-25):*

*"Overall this study shows, that skilful predictions of storm loss occurrences on lead times of several days can be made using the presented (fully probabilistic) framework to integrate meteorological forecast uncertainties and uncertainties resulting from impact model. Findings thus demonstrate their potential use in risk based warning systems."*

*is expanded to address the reviewers point(s):*

*"Overall this study shows, that skillful predictions of storm loss occurrences on lead times of several days can be made using the presented (fully probabilistic) framework to integrate meteorological forecast uncertainties and uncertainties resulting from impact model. Such quantification of both, potential impacts of severe weather and their respective likelihood forms the basis for developing risk based warning systems. By quantifying impacts and their likelihood, which is particularly relevant to recipients, the acceptance of weather warnings might be strongly enhanced. As one of the first national weather services, the UK Met Office has recently moved on to a risk based warning system (Neal et al., 2013). The basis of such warning system is formed by the risk matrix, composed of the two dimensions impact and likelihood. By quantification of both these dimensions, the presented framework can thus directly feed into such a*

*warning system."*

*Reference: Neal, R. A.; Boyle, P.; Grahame, N.; Mylne, K. Sharpe, M. Ensemble based first guess support towards a risk-based severe weather warning service Meteorological Applications, John Wiley Sons, Ltd, 2013*

**Technical corrections**

Generally: Define when talking about mean wind speeds and wind gust speeds. For wind impacts and studies using wind data knowing the difference is important.

Ensure paragraphs aren't one sentence long.

Keep consistent phrasing e.g. 'damage uncertainty' not 'impact model uncertainty' (e.g. page 2 line 14); 6-hourly (page 4 line 1) and 6 hourly (page 3 line 16).

Be consistent in the use of 'damage modelling uncertainty', 'meteorological uncertainty' and 'full uncertainty/both uncertainties' e.g. Page 9 line 1 it's referred to as 'damage occurrence uncertainty' and page 9 line 5 vs Figure 3 and Figure 4.

Ensure consistency of Brier Score and Brier Skill Score capitalisation or define then use BS and BSS don't mix.

Consistency needed for 'winter storm Britta' references including removing capitalisations.

Change all references of 1d, 3d and 9d to 1 day, 3 days and 9 days respectively including in figures.

Keep 'low' and 'high' threshold wording consistent, 'lower' and 'higher' thresholds are referred to on page 9.

Check references in particular comma placements and journal titles.

Page 1 line 14: clarify what are the 'two individual contributions'

Page 1 line 15: 'forecast skill' of what?

Page 1 line 17: define 'district level'

Page 2 line 8: source 'of' uncertainty not 'for'

Page 2 line 17: approach 'when' predicting

Page 2 line 26: comprise 'of' daily data

Page 3 line 2: 'insurers' not 'insurances'?

Page 3 line 15: define DWD

Page 3 line 16: 10 meter's' above 'the' ground

Page 3 line 18/19: 'hourly 10m wind gusts' instead of 'hourly wind gusts in 10m height'

Page 3 line 20: 'maximum 10m wind gusts' instead of 'maxima of wind gusts in 10m height'

Page 3 line 22: ECMWF 'has' operationally 'run' its

Page 3 line 24: can remove 'the'

Page 3 line 25: check sentence structure and TL use

Page 3 line 28: 'In' December or re-arrange sentence

Page 4 line 1: remove space between 10 and m

Page 4 line 11: 436,905 not 436.905; comma needed after 'equation'

Page 4 line 15: TL not TL

Page 4 line 18: maximum 10m 'wind' gusts

Page 5 lines 6 and 7: Makes very little sense, please revise

Page 5 line 11: 'Post-processing' no 'A' needed.

Page 5 line 12: 'what is meant in the study by the term 'calibration'? please revise

Page 5 line 13: depending 'on' the, not 'of'

Page 5 line 25: What are CDF's?

Page 5 line 26: PDF already defined

Page 6 line 6: move comma to after district and remove from after derived

Page 6 line 18: no 'the' needed before Sec. 3.3

Page 8 line 8: 'range is only between'

Page 8 line 15: 'previously' not 'above'

Page 8 line 20: Don't need 'Considering' at the beginning of the sentence.

Page 8 line 28: add 'observed' to make 'the observed loss ratio'

Page 8 line 30: add 'only' after 'uncertainty' and clarify/reword the sentence.

Page 9 line 1: 'uncertainty only for winter storm Britta' or 'uncertainty in the case of winter storm Britta'

Page 9 lines 2-4: reorder to refer to both uncertainties then dressed ensembles, this fits better with the order of Figure 3.

Page 9 line 9: 'forecasts' not 'forecast'

Page 9 line 13: 'When averaged over' not 'In average over'

Page 9 line 19: '3.5), with the climatology as a reference forecast, it is confirmed that'

Page 9 line 23: 'For a lead time'

Page 9 line 30: define 'high impact events'

Page 9 line 30: 'Even for a lead time'

Page 10 lines 4 and 5 and Page 11 lines 8 and 17: not sure 'stronger' and 'strong' are

the correct words, please revise

Page 10 line 6: 'in the case'

Page 10 line 16: 'winter storms' instead of 'severe winds'

Page 10 line 24: 'In forecasting'

Page 10 line 30: 'too little ensemble'

Page 10 line 2: 'for short lead times'

Page 10 line 3: no comma needed after 'fact'; 'uncertainties are smaller at' no need for 'obviously'.

Page 10 line 7: 'situations'

Page 10 line 8: 'improvement in forecast'

Page 10 line 16: 'large lead times'

Page 10 line 23: 'Overall, this study shows, for this particular case study, that'

Figure 3: Units on right hand side, referred to in

Figure 4: Make scales the same so easier to compare. Caption could say ' Lead time dependent Brier Skill Score (BSS; employing climatology as the reference forecast) for events with loss ratio exceeding low threshold (0.0001o/oo) (left) and loss events with loss ratio exceeding high threshold (0.001o/oo) (right). Shown in black symbols are verification results for the four different set-ups, red triangles show verification results using the ensemble dressing post processing system.'

Figure 5: 'for loss ratio exceeding low threshold (0.0001o/oo)'

*Thank you for these technical corrections! We will take care to incorporate them into the manuscript.*

[Figure]

**Fig. 1.** Reliability diagrams for the forecasts with lead time 3 days for the high loss threshold (0.001‰).

---

## Author Comment (AC2) · 6 Oct 2016

**Reply to RC2**

The authors would like to thank the reviewer for the very thoughtful review. The comments, suggestions and numerous technical corrections will certainly help us improve the manuscript. Point-by-point answers (in *italics*) to the comments are given in the following.

**General comments:**

The paper is clearly written and logically organised, it reads very well and the methodologies are well explained. Apart from one more general comment that can hopefully be addressed easily, I mainly found a couple of wording issues or typos that need to

be corrected. After some minor revisions the paper should be suitable for publication.

On the example of one storm (pages 8/9) the authors demonstrate how accounting for the uncertainties increases the probability of losses occurring, and this improves the forecast for the case when a storm / damage occurs as the deterministic model strongly underpredicted the losses. However, I am wondering if this also increases (and by how much) the probability of damage when no storm occurred, thereby leading to a higher false-alarm rate? I would appreciate if the authors could expand on issues around false alarm rate.

*This is a very good comment. We will add a discussion on false alarms to the manuscript.*

*In case of the deterministic forecasts (no uncertainty treatment), the hit rate (H=100% x hits / (hits + misses) ) and false alarm rate (FAR=100% x false alarms / (false alarms + correct rejects)) can directly be calculated from the contingency table. In case of probability forecasts, a threshold needs to be chosen to translate them into a deterministic one to be able to calculate FAR and H. This threshold can be freely chosen and strongly influences FAR and H. Naturally, trying to reduce the FAR will also reduce H and vice versa. Insight into this relation can be gained by assessing the ROC (Relative operating characteristic) curves, relating the false alarm rate (FAR) to the hit rate (H), depending on the probability threshold chosen. Compare Wilks (2008), Chapter 8 for details on the ROC curve.*

*We will add a figure (see Fig 1.) to the manuscript, showing exemplarily resulting ROC curves for a lead time of 3 days.*

*It can be clearly found, that using the probabilistic forecasts, the hit rate (H) can be strongly increased with only slight increase in the false alarm rate (FAR). Exemplarily, when considering the deterministic forecasts for a lead time of 3 days the hit rate is 3.5% (of all observed events, only 3.5% are forecasted), while the false alarm rate is 0.004% (an event was forecasted in only 0.004% of the cases for which no event was*

*observed) for the high threshold. By using the probabilistic forecasts, a much higher hit rate of 80% can be achieved while keeping the false alarm rate below 10%. In this way at least 80% of all events are correctly forecasted, which poses a great improvement, particularly since dealing with severe and damaging events.*

**Specific comments:**

Title: the paper does not only report on uncertainties but also quantifies skill and shows how treating uncertainties increases the skill. Therefore I think it would be good to add "skill" to the title, e.g. "Uncertainties and skill in forecasts: : :" or similar

*New title based on both reviewers' comments:*

*"An Analysis of Uncertainties and Skill in Forecasts of Winter Storm Losses"*

Section 2: As the different data (loss, COSMO, ECMWF EPS) cover different time periods, it would be good to say explicitly for which period the skill calculations were performed, for which period the downscaling was trained, etc.

*Details on the availability of individual data sources are already given in the data section. However we also corrected a mistake, namely that the dataset on losses on residential buildings is available for 1997-2011. We will add the information on the respective periods for which training and verification is performed.*

*"According to the data availability, the different modelling steps described in the following chapter are performed for different time periods. The statistical downscaling (compare Section 3.1) is developed on the basis of a set of simulations for individual storm events during the period 1959-2010. The ensemble post-processing (compare Section 3.2) is performed for the years 2006-2009 for which both COSMO-EU analyses and ECMWF-forecasts are available. The training of the probabilistic damage model (compare Section 3.3) can be performed for the years 2006-2011 for which both damage data and COSMO-EU analyses are available. Assessment of forecast skill is done for the period 2001-2009 for which ECMWF-forecasts and damage data are available."*

**Technical corrections:**

page 1, line 30: not all references seem to be in the reference list at the end of the paper

page 2, line 27: would "records" be more appropriate then "measurements" when talking about the insurance data?

page 3, line 4: typo: inhomogeneities

page 5, line 7: end of sentence behind "over-dispersion"

page 5, line 12: comma at end of line behind "forecast ensemble"?

page 5, line 28: representative of

page 6, line 9: split the long sentence behind "loss ratio time series. The resulting: : :"

page 6, lines 19/26: past tense of forecast is forecast

page 6 section 3.4, it seems there is no explicit reference to setup ii)

page 6, line 28: 0.5 (decimal point rather than comma)

page 7, line 5: "ordered according to" rather than "after"

page 7, line 16: should the "and" (after has occurred) better be "or"?

page 9, lines 2/4: consider replacing "featured" by "recorded"

page 9, line 21: do you show "significant" or rather "positive" forecast skill?

page 10, line 5: consistent with

page 10, line 9: may the higher skill in northern regions also be due to higher average loss ratios in these districts?

Section 5: most of this sections reads more like a "Summary" rather than "Conclusions and Discussion". I figure that the very last 3 lines (page 11, lines 23-25) may be your

main conclusion?

page 10, line 28: specify "meteorological analyses"

page 10, line 30: too little ensemble spread

page 11, line 3: remove comma behind "fact"

*Thank you for these technical corrections! We will take care to incorporate them into the manuscript.*
* * *
Interactive
comment

**Fig. 1.** ROC curves for the forecasts with lead time 3 days for the high loss threshold (0.001‰).